# Sex-Specific Control of Muscle Mass: Elevated IGFBP Proteolysis and Reductions of IGF-1 Levels Are Associated with Substantial Loss of Carcass Weight in Male DU6PxIGFBP-2 Transgenic Mice

**DOI:** 10.3390/cells9102174

**Published:** 2020-09-26

**Authors:** Daniela Ohde, Michael Walz, Christina Walz, Antonia Noce, Julia Brenmoehl, Martina Langhammer, Andreas Hoeflich

**Affiliations:** 1Institute of Genome Biology, Leibniz-Institute for Farm Animal Biology (FBN), 18196 Dummerstorf, Germany; ohde@fbn-dummerstorf.de (D.O.); walz.michael@fbn-dummerstorf.de (M.W.); walz@fbn-dummerstorf.de (C.W.); noce@fbn-dummerstorf.de (A.N.); brenmoehl@fbn-dummerstorf.de (J.B.); 2Institute of Genetics and Biometry, Leibniz-Institute for Farm Animal Biology (FBN), 18196 Dummerstorf, Germany; martina.langhammer@fbn-dummerstorf.de

**Keywords:** IGFBP-2, sex, transgenic mouse, phenotype selection, muscle accretion, carcass weight, IGF-1, proteolytic activity

## Abstract

In farmed animals, carcass weight represents an important economic trait. Since we had demonstrated that IGFBP-2 represents a potent inhibitor of muscle accretion in inbred mice, we wanted to quantify the inhibitory effects of IGFBP-2 under conditions of elevated protein mass in growth selected non-inbred mice (DU6P). Therefore, we crossed male DU6P mice with female IGFBP-2 transgenic mice. Male IGFBP-2 transgenic offspring (DU6P/IGFBP-2) were characterized by more than 20% reductions of carcass mass compared to male non-transgenic littermates. The carcass mass in males was also significantly lower (*p* < 0.001) than in transgenic female DU6P/IGFBP-2 mice, which showed a reduction of less than 10% (*p* < 0.05) compared to non-transgenic female DU6P/IGFBP-2 mice. Although transgene expression was elevated in the muscle of both sexes (*p* < 0.001), serum levels were normal in female, but significantly reduced in male transgenic DU6P/IGFBP-2 mice (*p* < 0.001). In this group, also IGFBP-3 and IGFBP-4 were significantly reduced in the circulation (*p* < 0.01). Particularly in male transgenic mice, we were able to identify proteolytic activity against recombinant IGFBP-2 included in diluted serum. IGFBP-proteolysis in males correlated with massive reductions of IGF-1 in serum samples and the presence of elevated levels of IGFBP-2 fragments. From our data, we conclude that elevated tissue expression of IGFBP-2 is an essential effector of muscle accretion and may block more than 20% of carcass mass. However, in the circulation, intact IGFBP-2 contained no reliable biomarker content. Notably, for the estimation of breeding values in meat-producing animal species, monitoring of IGFBP-2 expression in muscle appears to be supported by the present study in a model system.

## 1. Introduction

IGFBP-2 was identified as a candidate gene for negative growth regulation under conditions of growth selection [1]. In fact, forced expression of IGFBP-2 in transgenic mice confirmed IGFBP-2 as an inhibitor of somatic growth but further identified muscle as a particularly sensitive tissue for the negative effects of IGFBP-2 [2,3]. On a mixed inbred and outbred background (50% C57BL/6 and 50% NMRI), overexpression of IGFBP-2 under control of the cytomegalovirus promoter correlated with altered muscle metabolism defined by an increased number of glycolytic fibers and elevated activity of lactate dehydrogenase in male IGFBP-2 transgenic mice [4]. In Musculus rectus femoris from this model, it was further demonstrated that IGFBP-2 decreased myogenic proliferation and thus blocks muscle accretion [4]. In the same muscle from male IGFBP-2 transgenic mice, the weight reductions were further characterized by reduced cross-sectional area, whereas the number of myofibers was not affected by the genotype [4]. Notably, this effect was not identified in female IGFBP-2 transgenic mice. Additionally, local injection of adeno-associated viral vectors containing a CMV-IGFBP-2 expression construct in the anterior tibialis muscle reduced muscle mass and further reduced muscle force at the site of administration [5]. Accordingly, based on mouse models, it is evident that IGFBP-2 controls growth, metabolism, and function of the muscle. 

Several studies addressed the biomarker potential of IGFBP-2 for muscle accretion in vertebrates. Accordingly, in Angus bulls and in conjunction with parameters of growth hormone secretion, concentrations of circulating intact IGFBP-2 were informative for the prediction of average daily weight gain [6]. On the genomic level, studies in pigs [7,8] and chicken [9,10,11] have identified single nucleotide polymorphisms of the gene encoding IGFBP-2 associated with altered growth and carcass weight, and the inclusion of the IGFBP-2 locus in selection programs has been discussed. The negative effect of IGFBP-2 on somatic growth seems evolutionary conserved and was provided in zebrafish characterized by a duplicated *IGFBP-2* gene [12]. Just recently, a negative correlation between hepatic IGFBP-2 expression and growth rates was also described in fish breeds [13]. Finally, in humans, higher concentrations of IGFBP-2 in serum were correlated with reduced muscle strength and reduced physical performance and overall with a higher degree of disability in elderly male subjects [14]. Due to markedly different IGFBP-2 effects [15] or concentrations [16] in males and females, IGFBP-2 was discussed as a mediator of sex differences [17]. In fact, IGFBP-2 is regulated not only by growth hormone [18] but to a significant extent also by sex steroids [19]. Under conditions of malignant growth, IGFBP-2 can increase estrogen receptor expression and the activity of the phosphatidylinositol 3-kinase (PI3K) as provided by Foulstone et al. [20], suggesting an intricate relationship between sex steroid effects and IGFBP-2. In the brain of IGFBP-2 transgenic female but not in male mice, protein kinase B (AKT) phosphorylation was elevated if compared to controls [21], suggesting sex-specific regulation of PI3K also in vivo. Therefore, the study of sex differences for the pleiotropic effects of IGFBP-2 [21] and as performed here for the IGFBP-2 effect on the level of muscle accretion is mandatory. 

As a model for elevated muscle accretion in farm animals, we used the non-inbred DU6P mice, long-term selected for high protein amount [22,23]. We asked to which extent IGFBP-2 could block muscle accretion under conditions of high protein mass. Therefore, we crossed male DU6P mice with female IGFBP-2 transgenic mice [2].

## 2. Materials and Methods

### 2.1. Animal Production, Husbandry, and Tissue Sampling

Male mice from the non-inbred DU6P long-term selected mouse line were mated with female hemizygous IGFBP-2 transgenic mice [2] (C57BL/6; Figure 1). In IGFBP-2 transgenic mice, mouse IGFBP-2 is expressed under control of the cytomegalovirus (CMV) promoter in multiple tissues [2].

The study was approved by the National Animal Protection Board Mecklenburg-Vorpommern (file number: LALLF M-V/TSD/7221.3-1.2-037/06) and fully adhered to national and international laws. Offspring was standardized to ten animals (five male, five female) per litter immediately after birth and genotyped for the presence of the IGFBP-2 transgene in genomic DNA as described before [2]. Animals were kept at 12 h dark/12 h light cycles under conventional conditions and were fed with standard chow (Altromin 1314: protein, 22.5%; fat, 5%; raw fiber, 4.5%; ash, 6.5%; Altromin GmbH, Lage, Germany) and water ad libitum. Bodyweights were recorded at days 21, 35, and 49. At day 49, animals were sacrificed, serum was collected, and tissues were weighed, frozen in liquid nitrogen, and stored at −80 °C.

### 2.2. Protein Analysis by Western Blot

Tissue samples were homogenized as described before [24] in lysis buffer (New England Biolabs, Frankfurt, Germany) supplemented with protease inhibitors (cOmplete™ Mini; Roche, Mannheim, Germany) using the Precellys^®^24 (Peqlab Biotechnologie, Erlangen, Germany). Laemmli sample buffer containing beta-Mercaptoethanol was included, and samples were denatured at 94 °C. Cell debris was separated by centrifugation (10,000× *g*, 2 min, 4 °C) and 20–40 μg of protein from the samples were separated on 12% SDS-PAGE gels and transferred to carrier membranes (PVDF membrane, Millipore, Eschborn, Germany). Protein abundance and protein phosphorylation was investigated using specific antibodies also as described before (Table 1) [4].

### 2.3. Expression of Functional IGFBPs in Serum by Western Ligand Blotting

IGFBP-2, -3, and -4 were identified and quantitated by Western ligand blot (WLB) analysis using recombinant IGFBP standards as described previously [25]. In brief, all samples were denatured in sample buffer for 5 min. After gel separation (12% SDS-PAGE) and transfer to membranes (PVDF, Millipore, Bedford, MA, USA), all blots were incubated with the ligand (biotin-labeled human IGF-2, IBT-Systems, Binzwangen, Germany) and streptavidin-conjugated secondary antibodies (IBT-Systems, Binzwangen, Germany). Detection was achieved using enhanced chemiluminescence (Luminata^™^ Forte, Millipore, Bedford, MA, USA).

### 2.4. Measurement of IGF-1 and -2 in Tissue and Serum

Levels of IGF-1 and IGF-2 were assessed in muscle, liver, and serum by using an enzyme-linked immunosorbent assay (ELISA, Mediagnost, Germany) according to the manufacturer′s instructions. Tissues were homogenized in buffer with Precellys Ceramics Kit (Peqlab Biotechnologie, Erlangen, Germany), and calculated as ng IGF per mg protein.

### 2.5. Proteolytic Assay

Proteolytic activity against recombinant mouse IGFBP-2 (rmIGFBP-2) was assayed in serum samples derived from all experimental groups. Therefore, serum was dissolved in sterile PBS (1/10) and incubated at 39 °C in a thermoshaker (40 rpm) in the presence of 5 ng/µL rmIGFBP-2. As a control, proteolytic activity was assessed in artificial serum matrix (Biopanda, County Down, UK). Samples were taken at five consecutive time points (T = 0, 30, 90, 180, and 360 min) and mixed with an equal volume of 2-fold concentrated Laemmli buffer. IGFBP-2 was detected by WLB, as described above, but the signal intensities (AU) were expressed relative to the original signal intensity (relative quantification) at the start of the experiment (T0).

### 2.6. mRNA Expression of PAPP-As and STCs in Tissue

Total RNA from the liver was isolated and processed essentially as described before [24] with modifications as described here. The following primers were used: PAPP-A (pappalysin-1; [26]) forward 5′-TCCGCTCTTTCGACAACTTT-3′, reverse 5′-CATGGTAGTGGTGGTTGCTGG-3′; PAPP-A2 (pappalysin-2; [27]) forward 5′-ATTAATAACCGGGCCTACTGCAAC-3′, reverse 5′-GTCACAATCAGCAGCAAATGGAA-3′; STC1 (stanniocalcin 1; [26]) forward 5′-CCCAATCACTTC TCCAACAGA-3′, reverse 5′-GAAGAGGCTGGCCATGTTG-3′; STC2 (stanniocalcin 2; [28]) forward 5′-GAAATCCAGGGTTTACATGG-3′, reverse 5′-TCCTTGATGAATGACTTTCC-3′; housekeeping: Rpl19 (ribosomal protein L19) forward 5′-CAATGCCAACTCCCGTCAGC-3′, reverse 5′-TCTTGGATTCCCGGTATCTC-3′; Pgk1 (phosphoglycerate kinase 1) forward 5′-CAGTCTAGAGCT CCTGGAAGGT-3′, reverse 5′-GCCACTAGCTGAATCTTGCG-3′. The method of normalization was also described before [29].

### 2.7. Statistics

Analysis of body composition was achieved by the use of the SAS software package (version 9.3). For descriptive statistics and tests for normality, the procedure UNIVARIATE was applied. If the data were considered being normally distributed, the MIXED procedure (ANOVA) from the software package was applied. The analysis of variance considered sex (male and female) and genotype (non-transgenic, transgenic) as fixed factors. All other data were analyzed by one-way ANOVA (GraphPad Prism).

## 3. Results

### 3.1. Effect of IGFBP-2 Overexpression on Growth Parameters and Body Composition

At the age of 7 weeks, in male transgenic mice, a significant reduction of body weight by almost 12% (*p* < 0.001) was observed if compared to male non-transgenic littermates (Figure 2). In females, transgene expression resulted in no significant reduction of body weight compared to controls. The carcass weights were reduced by more than 20% in transgenic males (*p* < 0.001) and by almost 8% in transgenic females (*p* < 0.05).

The reductions of body and carcass weight were significantly stronger in male (*p* < 0.001) than in female transgenic mice. In isolated muscles, significant weight reductions were also present if transgenic mice were compared with sex-matched non-transgenic controls (*p* < 0.01). The inhibitory effects of transgene expression on body, carcass, and muscle weight were evident more clearly in male than in female mice. Inhibitory effects of IGFBP-2 overexpression were further present on brain weight in both sexes (*p* < 0.001). Interestingly, in females, the expression of the IGFBP-2 transgene led to significantly elevated epididymal and perirenal fat mass (*p* < 0.05). Particularly, epididymal fat mass was increased by more than 40% in female transgenic mice (*p* < 0.001) when compared to non-transgenic controls.

### 3.2. Effects of IGFBP-2 Overexpression in Isolated Muscle

Since we had observed sex effects on the body and carcass weight of IGFBP-2 overexpression, we asked if the tissue levels of IGFBP-2 are different in males and females, or could be responsible for the specific weight reductions in male and female mice. However, as demonstrated by Western immunoblotting, intact IGFBP-2 was present in tissue extracts from both sexes (Figure 3).

In samples from transgenic animals, an additional molecular weight form was present, characterized by higher molecular weight, which might be due to IGFBP-2 before cleavage of the leader sequence and thus intracellular IGFBP-2. The tissue levels of IGF-1 or IGF-2 were unaffected by the genotype and similar in muscle lysates from both sexes (Figure 4).

However, analysis of signal transduction revealed sex effects in muscle lysates (Figure 5) since reduced levels of total AKT, but elevated levels of phospho-AKT were found in transgenic females (*p* < 0.05). By contrast, the ratio of phosphorylated versus total AKT was reduced in male transgenic mice (*p* < 0.05). In addition, the levels of total ERK1/2 were reduced in female transgenic mice only (*p* < 0.01).

### 3.3. Effects of IGFBP-2 Transgene Expression in the Circulation

Since we could not identify different regulation of the IGF-system within the muscle, we assessed potential effects of IGFBP-2 transgene expression in the circulation. Surprisingly, IGFBP-2 was not elevated in male or female IGFBP-2 transgenic mice when compared to non-transgenic littermates (Figure 6). Instead, IGFBP-2 was reduced in male IGFBP-2 transgenic mice if compared to both male non-transgenic controls and female IGFBP-2 transgenic mice (*p* < 0.01). In addition, serum concentrations of IGFBP-3 and IGFBP-4 were lower in male IGFBP-2 transgenic mice than in non-transgenic or female transgenic littermates (*p* < 0.05). The reductions of IGFBP-2 in serum from female IGFBP-2 transgenic mice were not statistically significant (*p* = 0.16). However, increased amounts of IGFBP-2 fragments were observed in both transgenic IGFBP-2 groups with respect to non-transgenic littermates (*p* < 0.05; Figure 7). The levels of IGBP-2 fragments in male IGFBP-2 transgenic mice were higher compared to female IGFBP-2 transgenic mice (*p* < 0.05). The severe reductions of IGFBPs in serum from male IGFBP-2 transgenic mice further correlated with substantial reductions of IGF-1 in the circulation if compared to non-transgenic male or female IGFBP-2 transgenic littermates (*p* < 0.001; Figure 8).

In contrast to male non-transgenic controls, the concentrations of IGF-1 in male transgenic IGFBP-2 mice were reduced by ≈85%. The reductions of IGF-1 in serum from male IGFBP-2 transgenic mice were further correlated with significant reductions of IGF-1 concentrations in the liver of the same group compared to male non-transgenic or female IGFBP-2 transgenic littermates (Figure 8; *p* < 0.01).

### 3.4. Proteolytic Activity in the Circulation

From the reductions of intact IGFBPs and from the presence of high levels of IGFBP-2 fragments in serum, we had to assume sex-specific control of IGFBP-2 degradation in IGFBP-2 transgenic mice. In fact, if recombinant murine IGFBP-2 was incubated with diluted serum from male IGFBP-2 transgenic mice for 90, 180, or 360 min, intact IGFBP-2 was present at significantly lower levels compared to female IGFBP-2 transgenic or male non-transgenic littermates (Figure 9; *p* < 0.05). 

Proteolytic activity was also observed in non-transgenic male mice, due to reductions of IGFBP-2 after 180 and 360 min of incubation in comparison to 30 min only (*p* < 0.05). In addition, if recombinant IGFBP-3, -4, or -5 was incubated with diluted sera from male IGFBP-2 transgenic mice, almost complete degradation was observed (data not shown).

### 3.5. Expression of IGFBP-2 Proteases and Stanniocalcins in Muscle

In order to address the expression of the IGFBP-proteolytic system in the muscle, we studied muscular mRNA expression of PAPP-A/A2 and stanniocalcins 1 and 2 (Figure 10). Expression of PAPPAs and STCs clearly followed a sex-specific pattern with elevated levels of PAPP-A, STC1, and STC2 (*p* < 0.001) but reduced expression of PAPP-A2 in female mice of both genetic groups compared to male controls (*p* < 0.05). This approach could not demonstrate an effect of IGFBP-2 transgene expression on the proteolytic system in the muscle.

## 4. Discussion

In C57BL/6 inbred mice, we have demonstrated that IGFBP-2 is a potent inhibitor of carcass weight and muscle accretion [2,3]. Now we asked to which extent IGFBP-2 blocks muscle accretion under conditions of growth selection in mice. This question is particularly relevant for the estimation of the biomarker value of IGFBP-2 concentrations in serum from farmed animal species selected for meat production. Although, also in human subjects, IGFBP-2 is negatively associated with skeletal muscle mass, and it was speculated that IGFBP-2 might represent an indicator of musculoskeletal health and muscle strength [14,30].

In order to test specifically the effects of elevated IGFBP-2 expression on muscle mass, we crossed IGFBP-2 transgenic mice [2] with a non-inbred mouse model selected for high protein amount at the age of 42 days (DU6P). These animals are characterized by elevated muscle accretion and by a high growth phenotype [23,31]. The strategy to crossbreed a monogenetic inbred mouse model (IGFBP-2 transgenic mouse) with a non-inbred and phenotype-selected mouse model (DU6P) represents the combination of reverse genetics with classical genetics. This can be seen as a novel but certainly as an unusual approach of functional genome analysis in vivo.

The phenotype of the DU6P mouse model is based on multiple genetic events acquired or enriched during more than 135 cycles of selection (= generations) over more than 30 years at this time [32]. In fact, by this approach, we were able to demonstrate that overexpression of IGFBP-2 is capable of blocking up to 20% of carcass weight in a sex-specific manner since particularly male mice were highly sensitive to the inhibitory effects of IGFBP-2. The molecular mechanisms described here, in a functional context with growth inhibition, were, in part, surprisingly distinct from what we described in IGFBP-2 transgenic inbred mice [2].

Before we start with the description of specific differences in inbred mice (C57BL/6) and the model used here (DU6P/C57BL/6), we mention the parameters similarly affected in both models: first of all, overexpression of IGFBP-2 resulted in reduced somatic growth, both in inbred mice and non-inbred DU6P mice. Next, in both mouse models, carcass weight was identified as a sensitive parameter for the inhibitory effects in response to forced expression of IGFBP-2 [2,33]. Additionally, the brain weight was reduced, whereas fat mass was increased in both models [21]. 

However, in inbred mice, transgenic expression under control of the CMV promoter resulted in significant increases of IGFBP-2 in the circulation [2]. This finding was not observed in the mouse model used here, in which neither males nor females showed an increase in IGFBP-2 in the circulation. Instead, endogenous IGFBP-2 levels appeared to be suppressed in serum to a substantial extent under conditions of elevated IGFBP-2 expression in male IGFBP-2 transgenic DU6P/C57BL/6 mice. Since this group was characterized by more than 20% reduced carcass mass, we may conclude from this observation that serum IGFBP-2 concentrations have to be interpreted with care. In fact, divergent selection in mice also could come to this conclusion [34] since intact IGFBP-2 was elevated in serum from two mouse lines selected for low body weight (BEL and ROL) but was normal in a third small mouse model (MUL) when compared to their respective groups selected for high weight. In a fourth mouse model also selected for low body weight (EDL), serum concentrations of IGFBP-2 were lower than in the respective divergently selected high weight mouse line (EDH; [34]). 

Notably, not only the concentrations of intact IGFBP-2 were reduced, but also the concentrations of IGFBP-3 and -4 in the serum of male IGFBP-2 transgenic DU6P/C57BL/6 mice. The reductions of IGFBPs in serum from male IGFBP-2 transgenic mice correlated with high proteolytic activity as provided by the degradation of intact IGFBPs over time, on the one hand, and by the presence of IGFBP-2 fragments on the other. Here, we identified proteolytic activity against IGFBPs in male DU6P/C57BL/6 mice but not in inbred C57BL/6 mice (data not shown). Although controversially discussed [28,35,36,37,38,39], there is substantial evidence for an effect of growth hormone on the stability of IGFBPs in mice and men. Accordingly, in growth-selected mice (DU6P), characterized by elevated IGF-1 [40], elevated GH could impact on IGFBP-stability, whereas in C57BL/6 mice, with normal IGF-1 no such interference could be identified. An inhibitory effect of GH on the serum concentrations particularly of IGFBP-2 [41] is undisputed, and reduced IGFBP-2 concentrations in serum have been discussed even as biomarkers of GH-doping in athletes [18]. In addition to the effect of elevated GH, an effect of sex steroids may be involved on the control of IGFBP-stability, as demonstrated in hormone-replaced Turner patients [42]. Accordingly, future work will have to address the effects of GH and sex steroids on IGFBP-stability. A contribution of IGFBP-proteolysis was discussed [43] and demonstrated [44] for the control of somatic growth in mice and humans [45,46]. Remarkably, in hemofiltrate of human origin, 18 different fragments of IGFBP-2 were identified and characterized [47], revealing IGFBP2 fragmentation as a biologically diverse process with tremendous potential for biomarker research in the future. Additionally, in DU6P mice, we had discussed an active involvement of IGFBP-proteolysis for the acute regulation of somatic growth during the pubertal growth spurt and the regulation of IGF-bioactivity at peak growth periods before [40]. However, during the regulation of peak growth at the age of about four weeks, the reduction of IGFBPs in the circulation did not correlate with the suppression of IGF-1, as described here in adult mice. Furthermore, at younger ages, we could not observe any effect of sex [40], which is clearly evident in the present study. 

According to the current concept of the IGF-system [48,49,50,51], a proteolytic system is involved in the acute control of IGF-related normal and malignant growth. In this system, PAPP-A is the enzyme that cleaves IGFBP-4 and -5 [52] but also IGFBP-2 [50], whereas PAPP-A2 can proteolyze IGFBP-3 and 5 [53], in addition, also other enzymes are known to proteolyze IGFBPs [54]. Since the concentrations of PAPP-A are higher in male than in female human subjects, a sex effect can be assumed for the proteolytic control of the IGF-system [55]. Based on the proteolytic degradation of recombinant IGFBP2 over time and the concentration of IGFBP-2 fragments in serum, we postulate higher levels of IGFBP-proteolysis in males if compared to females. As a potential mechanism for sex-related control of IGFBP-proteolysis in mice, we have no direct evidence so far for PAPP-A, which is known to cleave IGFBP-2 [56], because PAPP-A mRNA expression in muscle was higher in females than in males. The activity of PAPP-A and -A2 is regulated by stanniocalcins 1 and 2 (STC1 and STC2; [57,58]). Since in muscle, mRNA expression of both inhibitors was higher in females than in males, the lack of proteolytic activity in serum from female mice might be related to sex-dependent differences of STC expression in muscle. However, to improve our understanding of the proteolytic degradation of IGFBPs, the role of age, sex, and expression of STCs or PAPP-As in tissues other than muscle must be addressed in future studies. STC2 has been demonstrated to coregulate signaling of AKT and ERK in colorectal cancer cells [59]. This finding also argues for a more detailed analysis of age- and sex-related STC or PAPPA expression. Nevertheless, this finding adds a novel candidate for the control of signal transduction in our animal model, since phosphorylation of AKT was increased in muscle from female IGFBP-2 transgenic DU6P mice. Interestingly, sex-specific activation of AKT was demonstrated also in brain lysates from female but not from male IGFBP-2 transgenic mice compared to matched controls [21], indicating a more general effect of sex on IGFBP-2 dependent AKT phosphorylation in mice. 

In response to forced expression of IGFBP-2 in DU6P mice, not only IGFBP-2 (transgenic and endogenous) but also other IGFBPs were proteolytically digested in male mice. Therefore, we may assume that in male mice, both PAPP-A and PAPP-A2 are active since IGFBP-2, -4, and -5 (cleaved by PAPP-A) and IGFBP-3 and -5 (cleaved by PAPP-A2) are degraded efficiently [56]. The reason for the more general reduction of IGFBPs in serum from male IGFBP-2 transgenic DU6P/C57BL/6 mice is unclear. However, as a consequence of excessive IGFBP-proteolysis and loss of most IGFBPs in the circulation of male IGFBP-2 transgenic mice, the IGF-binding capacity is severely reduced, potentially resulting in the substantial loss of IGF-1. Although at least part of the reductions of IGF-1 in male mice may also be due to reduced hepatic IGF-1 production since hepatic expression and secretion of IGF-1 represents a rich source of serum IGF-1 [60]. Accordingly, the negative effects of IGFBP-2 overexpression on carcass and muscle mass could be related to reduced IGF-1 concentrations in serum (systemic effect) and/or increased levels of intact IGFBP-2 in muscle tissues (local effect). On the tissue level, in muscle from male and female IGFBP-2 transgenic DU6P mice, intact IGFBP-2 was substantially increased compared to non-transgenic littermates. Since the levels of intact IGFBP-2 in muscles of male IGFBP-2-transgenic DU6P mice were higher, although not significant, compared to female IGFBP-2-transgenic littermates, we cannot exclude that the more significant reductions in carcass weights in males might also be related to altered local muscular transgene expression.

From our data, we conclude that elevated expression of IGFBP-2 efficiently impairs muscle accretion also under conditions of elevated protein accretion. In absolute terms, the expression of the transgenic IGFBP-2 in male DU6P/C57BL/6 mice resulted in a reduction of carcass weight as well as isolated muscle by more than 20%. However, the inhibitory effects of IGFBP-2 on muscle or carcass weight were not reflected by elevated concentrations of intact IGFBP-2 in the circulation, neither in males nor females. Only in males, weight reductions correlated with substantial reductions of IGFBP-2 together with IGF-1, IGFBP-3, and IGFBP-4 in serum and with significant IGFBP-proteolysis. Our study reveals biomarker potential of IGFBP-2 fragments for carcass weight in males and may recommend the inclusion of IGFBP-proteolysis in the biomarker panels of breeding programs designed for farm animal selection. If the loss of muscle mass in human subjects is monitored, the proteolysis of IGFBPs can also represent a novel content of biomarker information. Our results may have implications for IGFBP-biomarker research in vertebrates [6,7,8,9,10,11] since circulating concentrations of IGFBP-2 or other IGFBPs may not as a rule be interpreted in only one direction and clearly have a sex-related physiological background.

## Figures and Tables

**Figure 1 cells-09-02174-f001:**
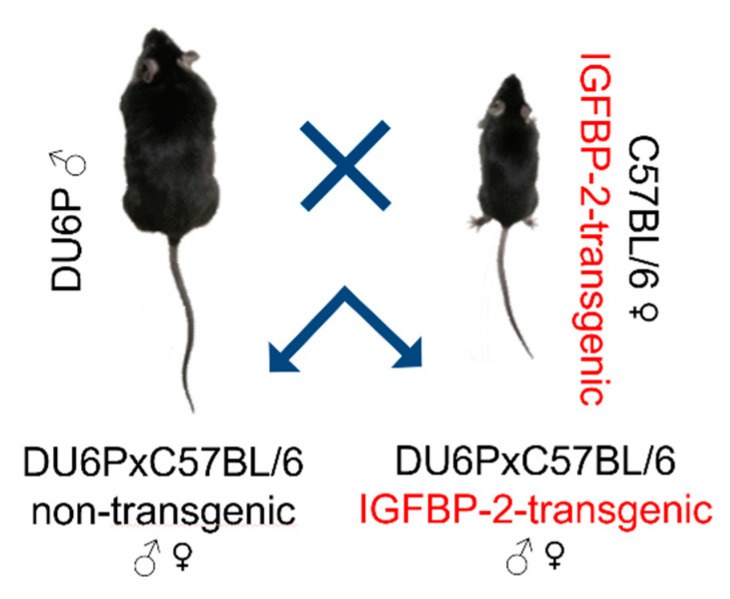
Breeding scheme: male mice selected for high protein mass (DU6P) were crossed with female hemizygous insulin-like growth factor binding protein 2 (IGFBP-2) transgenic mice (C57BL/6). Accordingly, the autosomal genetic background in the offspring was equally contributed by non-inbred DU6P mice and inbred C57BL/6 mice. Due to hemizygosity, the litters were composed of IGFBP-2 transgenic mice and non-transgenic controls.

**Figure 2 cells-09-02174-f002:**
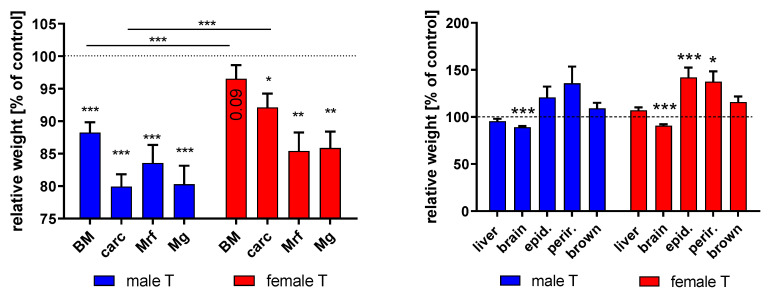
Body and organ weights in male and female IGFBP-2 transgenic mice (DU6P/C57BL/6) in percent of non-transgenic littermates at the age of 7 weeks (mean ± SEM; *n* = 14 for each group; *: *p* < 0.05; **: *p* < 0.01; ***: *p* < 0.001; BM: body mass; carc: carcass mass; Mrf: mass of Musculus rectus femoris; Mg: mass of Musculus gastrocnemius; epid.: epididymal fat mass; perir.: perirenal fat mass; brown: brown fat mass).

**Figure 3 cells-09-02174-f003:**
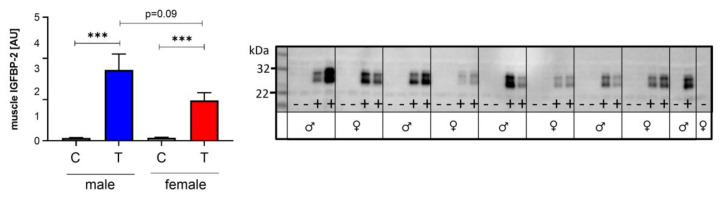
Expression of IGFBP-2 in muscle extracts from male (blue) and female (red) IGFBP-2 (T) and non-transgenic (C) littermates (DU6P/C57BL/6). Western immunoblotting was performed under reducing conditions. Immunoreactivity was present at two different molecular weights, which may represent IGFBP-2 with and without the C-terminal secretory signal peptide (mean ± SEM; age: 7 weeks; *n* = 9 for each group; +: IGFBP-2 transgenic; -: non-transgenic control; ***: *p* < 0.001).

**Figure 4 cells-09-02174-f004:**
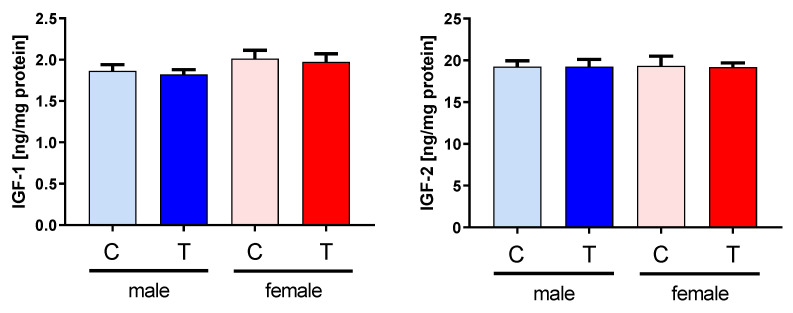
Concentrations of IGF-1 (left panel) and IGF-2 (right panel) in muscle extracts from male (blue) and female (red) IGFBP-2 transgenic (T) mice and non-transgenic (C) littermates (DU6P/C57BL/6). The concentrations of both peptide hormones were quantified by ELISAs as described in Section 2 (mean ± SEM; age: 7 weeks; *n* = 10).

**Figure 5 cells-09-02174-f005:**
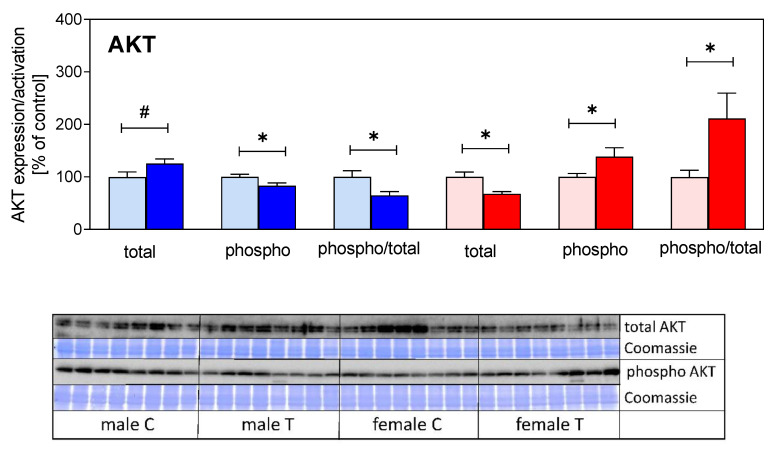
Protein expression (total), phosphorylation (phospho), and specific activity (phospo/total) of protein kinase B (AKT; upper panel) and extracellular regulated kinase (ERK1/2, lower panel) in muscle extracts from male (blue) and female (red) IGFBP-2 transgenic mice and non-transgenic littermates (DU6P/C57BL/6). Analysis of signal transduction was performed by Western immunoblotting as described in Section 2 (mean ± SEM; age: 7 weeks; *n* = 8; #: *p* < 0.1; *: *p* < 0.05; **: *p* < 0.01). The results from Western immunoblotting are provided both for AKT and ERK1/2. Coomassie blue staining was performed in order to demonstrate equal loading and protein transfer to the carrier membranes, respectively.

**Figure 6 cells-09-02174-f006:**
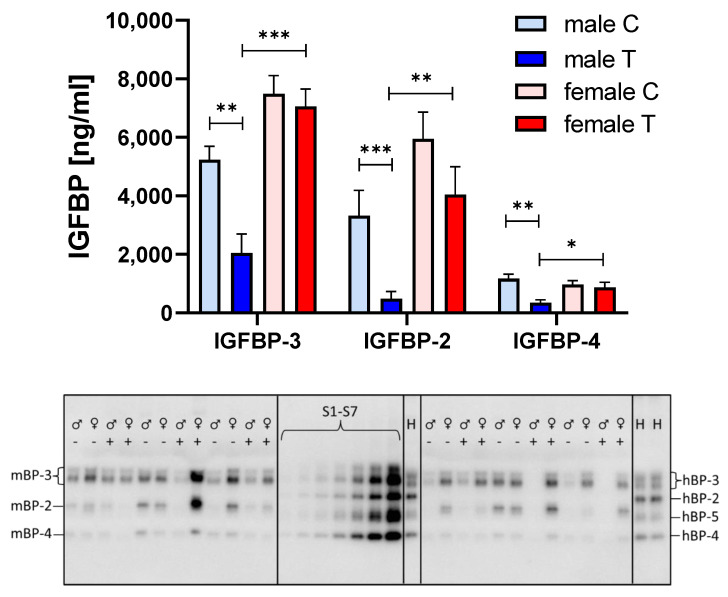
Concentrations of IGFBPs in serum from male (blue) and female (red) IGFBP-2 transgenic mice (T) and non-transgenic (C) littermates (DU6P/C57BL/6). Analysis was performed by quantitative Western ligand blotting (lower insert) as described in Section 2 (mean ± SEM; age: 7 weeks; *n* > 10; *: *p* < 0.05; **: *p* < 0.01; ***: *p* < 0.001; H: human control serum containing IGFBP-2 to -5; S1–S7: serial dilutions of human recombinant IGFBP-2 to -5 standards used for the quantification of IGFBPs on each blot).

**Figure 7 cells-09-02174-f007:**
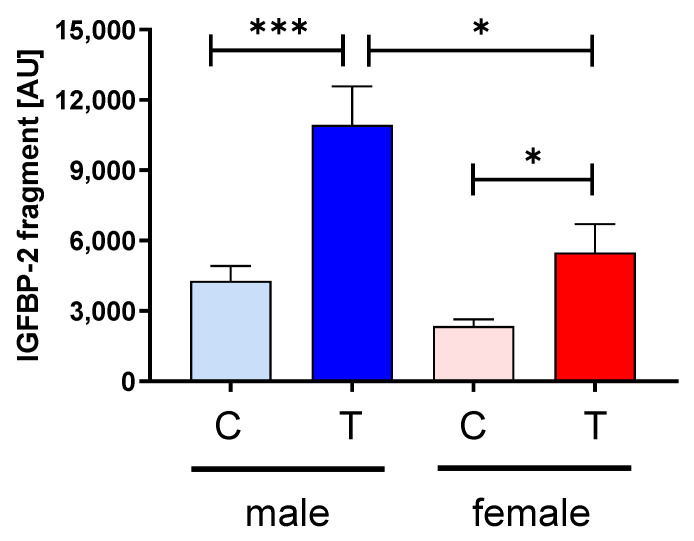
Relative quantification of IGFBP-2 fragments in serum from male (blue) and female (red) IGFBP-2 transgenic mice (T) and non-transgenic (C) littermates (DU6P/C57BL/6). Analysis was performed by Western immunoblotting as described in Section 2 (mean ± SEM; age 7 weeks; *n* > 7; *: *p* < 0.05; ***: *p* < 0.001).

**Figure 8 cells-09-02174-f008:**
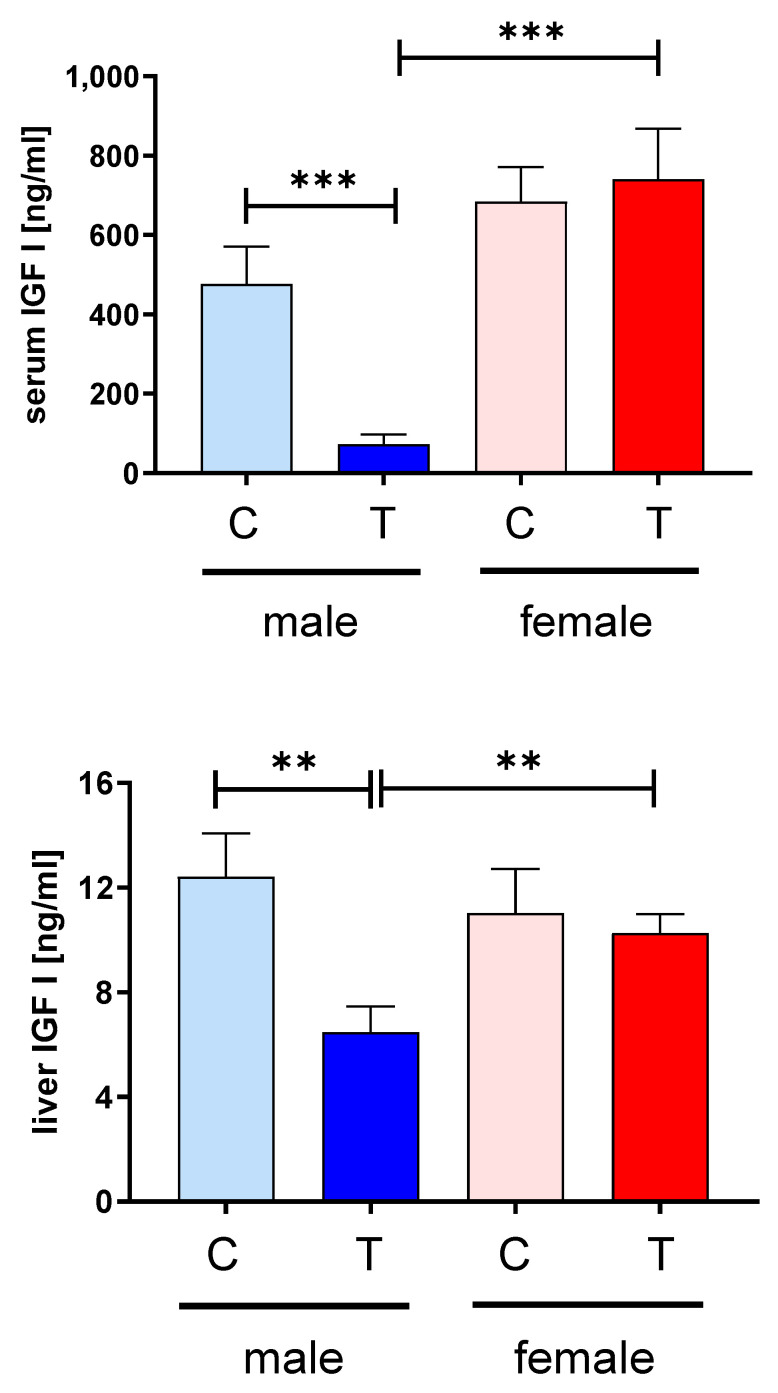
Concentrations of IGF-1 in serum (left panel) and the liver (right panel) from male (blue) and female (red) IGFBP-2 transgenic mice (T) and non-transgenic (C) littermates (DU6P/C57BL/6). Analysis was performed by ELISA as described in Section 2 (mean ± SEM; age: 7 weeks; *n* > 9; **: *p* < 0.01; ***: *p* < 0.001).

**Figure 9 cells-09-02174-f009:**
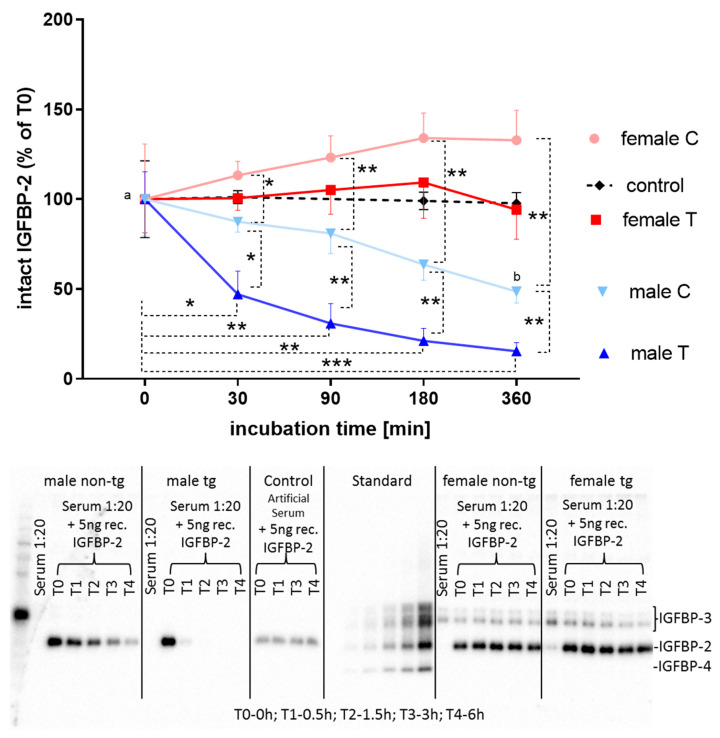
Stability of supplemented intact IGFBP-2 in serum from male (blue) and female (red) IGFBP-2 transgenic mice (T) and non-transgenic (C) littermates (DU6P/C57BL/6) as a function of incubation time at 37 °C. Diluted serum samples were supplemented with 5 ng murine recombinant IGFBP-2 for up to 360 min. The presence of intact IGFBP-2 was assessed by Western ligand blotting as described in Section 2 (mean ± SEM; age: 7 weeks; *n* > 5; *: *p* < 0.05; **: *p* < 0.001; ***: *p* < 0.001; a versus b: significant reduction of intact IGFBP-2 also in non-transgenic male mice after 360 min). As a negative control, artificial serum matrix was supplemented with 5 ng murine recombinant IGFBP-2 for the indicated durations (*n* = 3).

**Figure 10 cells-09-02174-f010:**
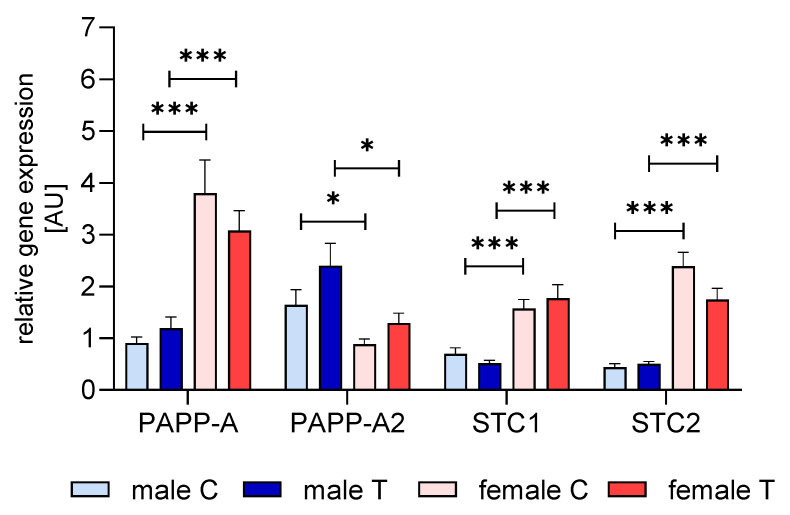
Gene expression of IGFBP-proteases (PAPP-A and PAPP-A2) and stanniocalcins (STC1 and STC2) in muscle from male (blue) and female (red) IGFBP-2 transgenic mice (T) and non-transgenic (C) littermates (DU6P/C57BL/6). The mRNA levels were assessed by quantitative real-time PCR and normalized for the expression of housekeeping genes as described in Section 2 (mean ± SEM; age: 7 weeks; *n* = 10; *: *p* < 0.05; ***: *p* < 0.001).

**Table 1 cells-09-02174-t001:** Antibodies used for the analysis of signal transduction in muscle tissue.

Antibody	Source
AKT	Cell Signaling (#9272)
Phospho-AKT (Ser473)	Cell Signaling (#4060)
p44/p42 MAPK (ERK1/2)	Cell Signaling (#4695)
Phospho-p44/p42 MAPK (ERK1/2) (Thr202/Tyr204)	Cell Signaling (#4377)
IGFBP-2	Santa Cruz Biotech (#sc-6002)

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
