# Peer review of "Sex-Specific Control of Muscle Mass: Elevated IGFBP Proteolysis and Reductions of IGF-1 Levels Are Associated with Substantial Loss of Carcass Weight in Male DU6PxIGFBP-2 Transgenic Mice"

_cells, 2020, doi:10.3390/cells9102174_

Round 1

Reviewer 1 Report

This is a well written and presented study of the gender-specific control of muscle mass in response to IGFBP-2 overexpression in growth-selected non-inbred DU6P mice. The comparisons with in-bred mice overexpressing IGFBP-2 was especially appreciated.

Specific Comments/questions:

  1. Were DU6P mice characterized for GH secretion? Alterations in GH could factor into interpretation of some of the results.
  2. Fig 5B. Relative gene expression (fold increase) - compared to what?
  3. In the Discussion it should be clarified that PAPP-A and PAPP-A2 are not the only IGFBP proteases. Granted, PAPP-A is the only protease for IGFBP-4, but several proteases can have IGFBP-3 and -5 (and IGFBP-2?) as substrates.

Author Response

Reviewer 1: This is a well written and presented study of the gender-specific control of muscle mass in response to IGFBP-2 overexpression in growth-selected non-inbred DU6P mice. The comparisons with in-bred mice overexpressing IGFBP-2 was especially appreciated.

Author response: We want to thank for the review of our work, positive response, and important input.

Reviewer 1: 1. Were DU6P mice characterized for GH secretion? Alterations in GH could factor into interpretation of some of the results.

Author response: We did not measure GH-secretion directly but measured IGF-1, which is substantially increased in DU6P mice as published recently (Walz et al, 2020 Cells). Accordingly, elevated GH in fact can be discussed in a context of IGFBP-stability and serum levels in our model. We are extremely grateful for this input because this will guide future work on the model! We are happy to extend the discussion of the revised manuscript according to the Reviewers suggestion (lines 329ff).

Reviewer 1: 2. Fig 5B. Relative gene expression (fold increase) - compared to what?

Author response: This in an error from our side. It must read: “relative gene expression [AU]”. We calculate gene expression relative to a standard series and normalize to a software-validated set of housekeeping genes. We have made corrections in the axis, accordingly (line 276).

Reviewer 1: 3. In the Discussion it should be clarified that PAPP-A and PAPP-A2 are not the only IGFBP proteases. Granted, PAPP-A is the only protease for IGFBP-4, but several proteases can have IGFBP-3 and -5 (and IGFBP-2?) as substrates.

Author response: Thank you for this clarification. We made changes according to the Reviewers suggestions (line 351).

Reviewer 2 Report

In this manuscript, Ohde and colleagues report on the inhibitory effects of IGFBP-2 under conditions of elevated protein mass in growth selected non-inbred mice. The study is overall sound, except for few points here reported:

80: The authors did not present how the transgenic mice were generated. It is important for better understand how and where IGFBP-2 is overexpressed

172: the authors analyzed the levels of IGFBP-2 in muscle to understand if different protein levels could explain the difference visible in body weight. Have the authors analyzed the expression levels of IGFBP-2 in the muscle of IGFBP-2 overexpressing mice, to verify if a difference in IGFBP-1 levels between genders could be a characteristic of these mice?

192: the authors showed only the quantification of western blot experiments. Representative images of western blot must be reported. It will be necessary also the presence of a reference gene (actin for example) useful to evaluate the amount of total protein presented.

245-246: the authors should modify the sentence since is not clear which sera were compared in the analysis of proteolytic activity. Moreover, how do the authors explain the increasing levels of intact IGFBP-2 in the female C group?

302: The authors showed that IGFBP-2 transgenic mice are characterized by the absence of increased levels of circulating IGFBP-1 compared to control animals. Since the upregulation of IGFBP-1 expression was expected, as visible in muscles, can the authors show the levels of IGFBP-1 into the principal organs to understand how the upregulation of the protein expression is visible? Since is not reported, is the expression of IGFBP-1 driven by a constitutive promoter or by a tissue-specific one?

319: IGFBP-1 transgenic mice are characterized by increased proteolytic activity that results in increased IGFBP-1 degradation. The authors declare the absence of this phenomenon in C57BL/6 control mice. How is the level of proteolytic activity against IGFBPs in DU6P mice? Is the upregulated expression of IGFBP-2 linked to an increase in its fragmentation?

Author Response

Reviewer 2: In this manuscript, Ohde and colleagues report on the inhibitory effects of IGFBP-2 under conditions of elevated protein mass in growth selected non-inbred mice. The study is overall sound, except for few points here reported.

Author response: We want to thank for critical revision of the manuscript bu also for the positive response and suggestions, which were all addressed in the revised manuscript.

Reviewer 2: 80: The authors did not present how the transgenic mice were generated. It is important for better understand how and where IGFBP-2 is overexpressed

Author response: We included specific information on the CMV-IGFBP-2 transgenic mouse model as requested (lines 80-81) .

Reviewer 2: 172: the authors analyzed the levels of IGFBP-2 in muscle to understand if different protein levels could explain the difference visible in body weight. Have the authors analyzed the expression levels of IGFBP-2 in the muscle of IGFBP-2 overexpressing mice, to verify if a difference in IGFBP-1 levels between genders could be a characteristic of these mice?

Author response: We studied tissue levels of IGFBP-2 in males and females in order to address the question if different levels of transgene expression may cause different inhibitory effect on the tissue level. We tried to make this point more clear in the revised manuscript (line 176-177).

Reviewer 2: 192: the authors showed only the quantification of western blot experiments. Representative images of western blot must be reported. It will be necessary also the presence of a reference gene (actin for example) useful to evaluate the amount of total protein presented.

Author response: We included images of the complete Western blots and of the reference band used for the demonstration of equal protein amounts on the membranes in the revised manuscript. We do not use actin because we and others found regulation of this protein in different systems. Therefore, we use more global markers, here derived by Coomassie staining of the membrane, and which clearly demonstrate if different amounts of proteins are present on the PVDF membranes. A currently frequently used global marker in protein biochemistry is provided by the Stain Free System distributed by BioRad. We provided the Coomassie stains for every gel in the revised manuscript (lines 199-213).

Reviewer 2: 245-246: the authors should modify the sentence since is not clear which sera were compared in the analysis of proteolytic activity. Moreover, how do the authors explain the increasing levels of intact IGFBP-2 in the female C group?

Author response: We made clear that male IGFBP-2 transgenic mice where compared. The increase of IGFBP-2 in the female C group is not significant. We may assume biological variance as a reason for the numerical differences of the means. However, in the manuscript we do not discuss non significant effects. Given the possibility, that an increase would be significant, we know that IGFBP-2 is present in higher molecular weight complexes, which may serve as reservoirs for free or monomeric intact IGFBP-2. Based on the results of the present manuscript, this discussion is not possible (lines 255-256).

Reviewer 2: 302: The authors showed that IGFBP-2 transgenic mice are characterized by the absence of increased levels of circulating IGFBP-1 compared to control animals. Since the upregulation of IGFBP-1 expression was expected, as visible in muscles, can the authors show the levels of IGFBP-1 into the principal organs to understand how the upregulation of the protein expression is visible? Since is not reported, is the expression of IGFBP-1 driven by a constitutive promoter or by a tissue-specific one?

Author response: We do not have the complete set of tissues in store in order to study transgene expression in all principal organs in both genders. Therefore, we cannot provide transgene expression in all tissues. However, transgene expression is driven by the CMV promoter, and with exception of the liver, IGFBP-2 expression can be expected and as a rule was found in all tissues studied so far in every IGFBP-2 transgenic mouse line. We have included this information on the promoter in the revised manuscript (line 312).

Reviewer 2: 319: IGFBP-1 transgenic mice are characterized by increased proteolytic activity that results in increased IGFBP-1 degradation. The authors declare the absence of this phenomenon in C57BL/6 control mice. How is the level of proteolytic activity against IGFBPs in DU6P mice? Is the upregulated expression of IGFBP-2 linked to an increase in its fragmentation?

Author response: Upregulation of IGFBP-2 is due to transgene expression. From the present data we do not have evidence to like expression and fragmentation. However, IGFBP-2 fragmentation was linked to the DU6P mouse model, which is characterized by high IGF-1 levels in males and females, as published before (Walz et al. 2020 Cells). By contrast, C57BL/6 mice have normal IGF-1 levels. Since also Reviewer 1 asked in a mechanistic  direction, we went back to the literature (as explicitly recommended by Reviewer 1) and found a really interesting potential mechanism, as GH is discussed in a context of IGFBP-stability. We therefore further developed the discussion in the revised manuscript (lines 329ff). We hope this amendment provides at least some support for a potential mechanism, which needs to be confirmed in future experiments:

 “Although controversially discussed [28, 35-39], there is substantial evidence for an effect of growth hormone on stability of IGFBPs in mice and men. Accordingly, in growth selected mice (DU6P), characterized by elevated IGF-1 [40], elevated GH could impact on IGFBP-stability, whereas C57BL/6 mice, with normal IGF-1 no such interference could not be identified. An inhibitory effect of GH on the serum concentrations particular of IGFBP-2 [41] is undisputed, and reduced IGFBP-2 concentrations in serum have been discussed even as biomarkers of GH-doping in athletes [18]. In addition to the effect of elevated GH, an effect of sex steroids may be involved in the control of IGFBP-stability as demonstrated in hormone-replaced Turner patients [42]. Accordingly, future work will have to address effects of GH and sex steroids on IGFBP-stability.”

Round 2

Reviewer 2 Report

The Authors answered to all criticisms.